# Open Drug Knowledge Graph

Mark Mann[1], Filip Ilievski[1], Mohammad Rostami[1], Aastha[2], and Basel Shbita[1]

[1] Information Sciences Institute, Marina del Rey, CA 90292, USA
{mbmann,ilievski,mrostami,shbita}@isi.edu
[2] University of Southern California, Los Angeles CA 90007, USA
aastha@usc.edu

**Abstract.** Automatic knowledge-based systems can assist medical professionals to make more informed recommendations and decisions. Unfortunately, as no comprehensive knowledge base (with both medical and non-medical) knowledge exists today, much manual effort is required to consolidate knowledge across sources, that are heterogeneous in content and formats. In this paper, we propose a knowledge-based method that aims to harmonize four such heterogeneous sources into a single drug-centric knowledge graph. The graph is based on the drugs found in Wikidata, and extended with specialized sources through an extraction and transformation pipeline, including data acquisition, entity resolution, and semantic modeling. Our analyses show that the resulting graph and its embeddings can capture drug similarity through their associated symptoms, and thus address common, knowledge-intensive medical search scenarios. As such, it holds the promise to be adapted for drug recommendation in the future. Given the modular setup of our method, new sources can be included to accommodate healthcare object use cases, relating to diagnoses and claims. We make the resulting knowledge source available in both relational database and property graph format.

## 1 Introduction

Healthcare systems heavily rely on the knowledge and the experiences of the physicians for drug prescription based on diagnosed symptoms of patients. Despite being dominant, this traditional process is limited to the knowledge scope of one person and faces several challenges.

1. Several different types of drugs may be appropriate to treat the same disease. In such situations, other (non-medical) factors such as price, accessibility, and insurance policy may help healthcare professions to reach optimal decisions.
2. Many healthcare professionals who are not physicians are not supposed to prescribe drugs in normal situations, yet they may need to act upon symptoms that they can diagnose in emergency situations to initiate treatment before an accurate examination can be performed by a doctor.
3. When a novel disease emerges, clinical data and standard treatment protocols are limited in the beginning, as in the case of COVID19. Physicians

may want to search for all potential existing drugs which may have positive effect given the observed symptoms of disease and then repurpose them for potential early stage treatment options [9].

4. Patients may desire to be more involved in the prescription process, e.g., knowing more about particular drugs and their side effect to improve the prescription process. Patients also may need to find the right drug at a reasonable price to purchase, in particular in the case of over-the-counter drugs.

Automated knowledge-based systems could assist with such tasks that involve intelligent searching of a database with the goal of arriving at a valid conclusion. We observe that, while a set of very valuable sources is publicly available, no comprehensive database exist that can accommodate the listed challenges. Existing medicinal drug databases, e.g., DrugBank[3], are helpful but they are mostly unstructured with abundant amount of thorough information about each entity, scattered across documents and not tailored to particular use cases. As a result, these sources are suboptimal for practicing medicine efficiently [11]. As a consequence, the user must spend a considerable amount of time to search across disjointed databases and narrow down to find the right treatment, but also to consider non-medical constraints such as price, and avoid adverse interactions with the current medications. For example, GoodRx[4] has structured drug prices and store availability for each medicine, but it can be only used for shopping after prescription as it lacks mapping of symptoms to drugs. WebMD[5] contains structured treatment data for each symptom but does not inform what over-the-counter drugs could help. DrugBank is an open-source database which can help to find which drugs are safe to consume with the current medications the patient is taking. However, the average person or even physician is not a computer scientist and cannot query this rich resource. Coming up with structured knowledge bases that integrate such existing distributed knowledge would help healthcare professionals to transcendent the above challenges and obtain accurate answers for their queries in a short time. It can also assist patients to buy drugs with better prices and improve their shopping experience.

In this paper, we develop a structured database for drugs in terms of a knowledge graph (KG) [8]. KGs have been found helpful in AI-aided medicine, in particular for clinical decision support systems for diagnosis and treatment [3, 14]. Building KGs using unstructured medical data helps performing more complex tasks using AI, including adverse drug reactions [2], drug discovery [10], repropose [9], and predicting drug-drug interaction [5]. Our goal is to construct a KG such that it can help the user to find potentially helpful drugs that can serve as potential treatments given a list of symptoms or a disease. Additionally, information about availability of drugs at nearby stores is provided to the user. Building upon the existing healthcare literature [17, 16], our goal is to integrate existing sources to create a comprehensive fast search experience for users who

---

[3] https://go.drugbank.com/releases/latest

[4] https://www.goodrx.com/

[5] https://www.webmd.com/drugs/2/conditions/index

manage conditions, budget, and control adverse drug interactions for patients. By integrating multiple knowledge sources, we enable the users to have more expressive search results in a short time. Our knowledge graph builds on the knowledge of symptoms to disease mapping. This helps to find possible drugs that can be used to treat a symptom. It incorporates information on prices and drug availability. This helps the user to zero down and research the drugs that are affordable and available.

We list the contributions of this paper as follows:

1. We present a pipeline for extraction and consolidation of relevant knowledge about symptoms, drugs, and their interaction, as well as non-medical information, such as drug prices. We apply our pipeline method to four relevant and complementary sources, resulting in an integrated knowledge base. **(Section 2)**
2. We make the resulting data publicly available, both in the form of a relational database and in the form of a knowledge graph.[6] The two formats support complementary use cases.
3. We analyze the contents of the resulting database. We provide statistics of its constituting nodes and relations, and run graph embedding-based queries to find similar products or drugs. **(Section 3)**
4. We assess the applicability of our integrated KG, by designing a user-friendly web interface and showing its utility on two representative scenarios. **(Section 4)**

## 2 Approach

The overall architecture of our approach is shown in Figure 1. We start by describing the data acquisition from the four sources that we will use in this paper: Wikidata [13], DrugBank, WebMD, and GoodRx (Section 2.1). We next describe their consolidation through entity linking and resolution between pairs of sources (Section 2.2). The resulting ontology of our data is described in Section 2.3.

### 2.1 Sources and data acquisition

We sought to construct a knowledge graph from several drug-centric data sources. Each source contributes with a particular set of information about drugs, their prices, and relations to conditions, which can be complimentary. To ease the effort required in entity linkage, we chose a well-adopted, drug-centric external id (Drugbank ID) as the primary key of our drug entity. For each data source, we identified the target features needed, and devised methods for extraction of the data:

1. **Wikidata** [13] is one of the largest publicly available knowledge graphs, describing over 90 million entities with more than a billion statements. To retrieve relevant data from Wikidata, we query it for medication (Q12140) entities with any Drugbank ID (P715) that treats any condition (P2175). We also

---

[6] https://www.kaggle.com/mannbrinson/open-drug-knowledge-graph

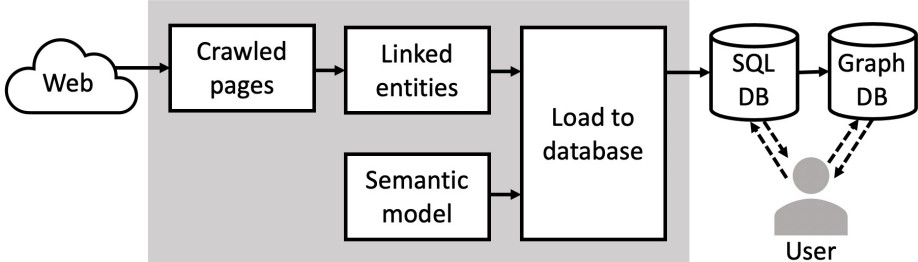

Fig. 1: Pipeline for Constructing Open Drug Knowledge Graphs. Each of the four data sources (Section 2.1) were extracted from the web. Then linked entities were created to join data sources. A semantic model was then used to format the data, and subsequently load into a relational and graph database.

    retrieve additional, optional features: the medication's active_ingredient_in (P3780), significant_drug_interaction (P769), and ATCCode (P267). The total amount of rows extracted from said query was 1,560.

2. **Drugbank** is a drug-centric database focused on drug-drug interactions and bioinformatics related features. Its knowledge is provided as data dump in XML format. We extracted all 2,166 drugs from Drugbank's XML dump, each with a maximum of 20 products and 100 interactions.

3. **WedMD** is a site focused on helping users search for treatments for a given condition. The site displays an index of all possible conditions, sorted alphabetically. From each condition, a list of drug treatments is provided. As the website provides no public API, we scraped its content programmatically. The crawler obtained a total of 58,921 condition-drug relations, and 12,857 unique drugs. Features extracted include: condition, product, user_reviews, and prescription_type.

4. **GoodRx** is a healthcare company that tracks prescription drug prices in the United States and provides free drug coupons for discounts on medications. As GoodRx does not provide a public API service, we extracted knowledge on GoodRx's drug products directly from their website, starting from a Wikidata-based seed list. The features of drug products that we extracted were: zipcode, store, price type, price, and price link. An total of 20,688 prices and 23 stores were extracted for the 997 matched drug products.

## 2.2   Entity resolution

The data extraction step is followed by an entity resolution step. As the entities across sources are originally disjoint, linking them is essential for the construction of a well-connected drug knowledge graph. To avoid introducing false positives, we first perform entity resolution across sources based on their external drug identifier (Drugbank ID). In this way, the Drugbank ID allowed us to link all

| Metric | Value |
|---|---|
| All Pairs | 178,519 |
| Pairs matched | 1,701 |
| Pairs found | 38 |
| Recall | 97.36 |
| Precision | 100 |
| True positives | 0.973 |
| False positives | 0 |
| True Negatives | 1 |
| False negatives | 0.026 |

Table 1: Wikidata to WebMD Entity resolution statistics. Entity linkage upon the 'drug product' feature between Wikidata and WebMD are reported in Table 1. A development set of 39 true pairs was used to evaluate the performance of our entity linkage method. Entity linkage method included blocking, feature bi-grams, and jaccard similarity threshold.

data sources to Wikidata in a 'hub-and-spoke' manner. This design choice enriched the information about the entities found in Wikidata, but excludes the remaining entities in the other three sources, which are not mapped to Wikidata through the Drugbank ID. For this purpose, we consider further linking on these entities. Specifically:

**Wikidata to Drugbank**: Linkage occurred only between Wikidata drugs (containing a Drugbank ID) and the subset of Drugbank drugs with matching Drugbank ID. In this case, we did not perform fuzzy matching, as we found it to decrease the overall quality of matching. Drugbank IDs were found on 787 wikidata medications.

**Wikidata to GoodRx**: We matched Wikidata and GoodRx based on exact matching query on the GoodRx website. URL requests to GoodRx return a result if the drug name is exactly matched, and otherwise give an 404 error. Of all 1560 wikidata drug products, we found 997 matches in GoodRx (recall of 63.9%).

**Wikidata to WebMD**: Due to absence of shared identifier between Wikidata and WebMD, we resorted to fuzzy matching between their drug products. A scoring function was leveraged to create matches for pairs, if the pair had a Jaccard similarity greater than 0.7. For each search term, bi-gram sets were generated before Jaccard similarity was calculated. A development set of 50 true pairs was manually compiled to enable evaluation of this matching approach. Hash-based blocking upon the entities first two characters was utilized to reduce candidate pairs from 20M to 178k. Our scoring function obtained 97.3% recall and 100% precision on these development pairs. Detailed results are shown in Table 1. We judge this level of error to be acceptable, thus, we proceed with this linkage strategy.

An overview of the number of entities mapped between Wikidata and each of the three other sources is shown in Figure 2. We allowed a one-to-one match for Wiki-Drugbank and Wiki-GoodRx matching tasks. However, we allowed one-to-many match for Wiki-WebMD drug product matching. This is because WebMD displayed many suffix variations for a product (ex: Adriamycin vial, Adriamycin-Pfs Solution) that we wanted to include in our graph. This design choice allowed for more matches (1701) than products existing in Wikidata (1560).

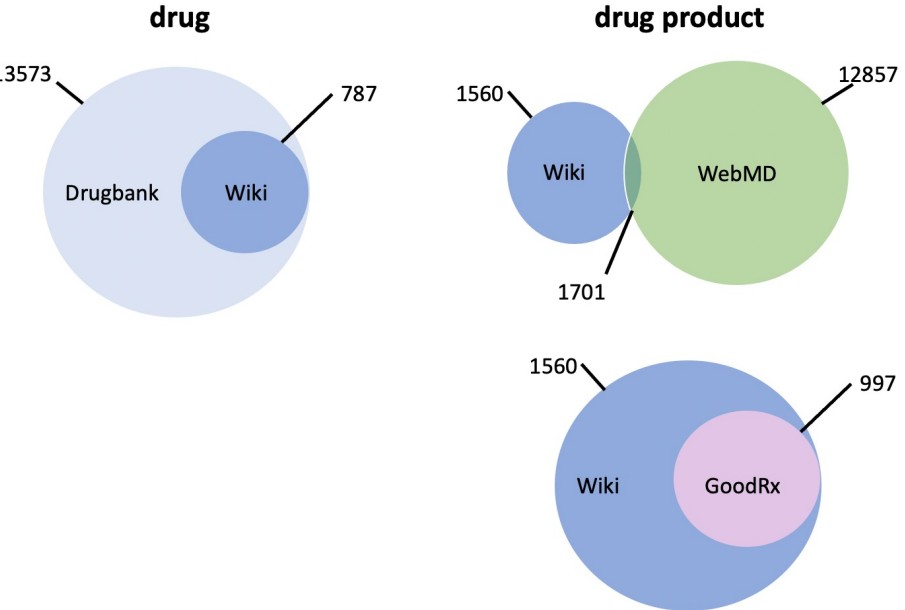

Fig. 2: Entity linkage metrics by data source. Displays the amount of entity intersections discovered between data sources. For Wiki-Drugbank linking, linkage occurred on the 'drug' entity using the Drugbank ID. For Wiki-WebMD and Wiki-GoodRx linking, linkage occurred on the 'drug product' entity using jaccard similarity and exact match methods, respectively.

### 2.3   Ontology design

The ontology was designed in a top-down manner, to fit our ultimate goal of enabling queries to connect patients with treatment based on their search parameters. We preserved all binary relations: treatment, interaction, active_ingredient_in, and drug_price, and used them to model information in all their suitable sources. To contain scope for our proof-of-concept, we selected these relations from Wikidata and Drugbank, while using all extracted relations from WebMD and GoodRx. We decided to categorize drug-like entities into two nodes - drug and product

- to represent the active ingredient and its name in the market. Our ontology map is described in figures 2 and 3. It is a simple yet powerful ontology, which allows us to achieve the project goals, including: (a) Store symptoms to drugs mapping (b) Capture drug interactions (c) Capture drug prices and variation across stores/zipcodes.

Figure 3 is a Entity-Relation diagram of the entities in our relational data model, created after entity linkage was completed. The relational data model was stored in a MySQL instance, and used as a back-end for our front-end application (Section 4.1). Figure 4 is composed of the same data, but expressed as a property graph and stored in Neo4j. The property graph format opened opportunities for us to leverage Neo4js robust graph-centric libraries for path-finding, centrality, and computation of embeddings.

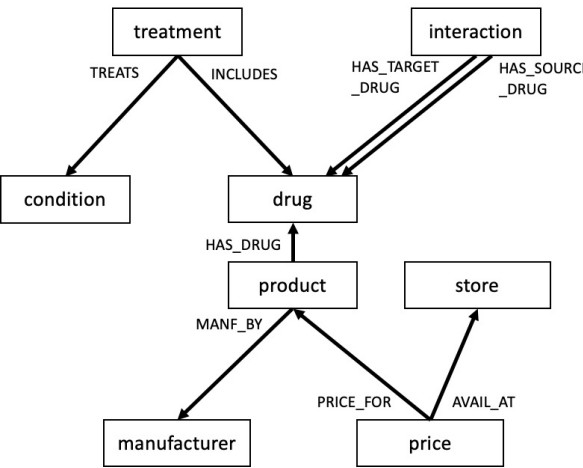

Fig. 3: Relational schema. Displays the semantic model in the format of an Entity-Relation (ER) diagram. In this model, each entity is a relational table in a MySQL instance.

## 2.4   Implementation

After extraction, and entity linkage scripts steps, the resulting data model was loaded to a MySQL instance using another python script. The relational schema is displayed in Figure 3. The resulting relational database was then exported in .csv format, and loaded to Neo4j. Neo4j import commands were utilized to load the data to create nodes and edges corresponding o the data model in Figure 4.

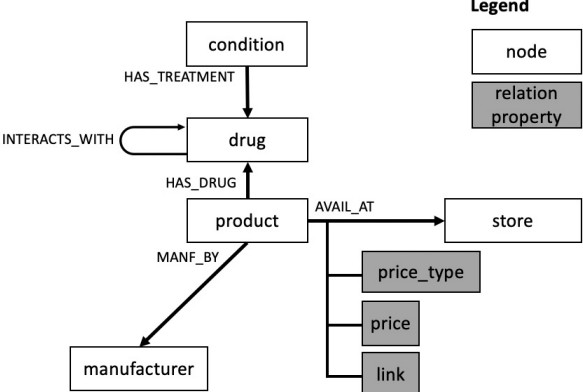

Fig. 4: Property graph schema. Displays the semantic model in the format of an property graph diagram. In this model, entities are represented as nodes and properties as edges. The property graph model allows for edge-centric properties, seen here on the 'AVAIL_AT' property.

## 3   Analysis

In this section, we analyze the contents of our knowledge base. First, we provide basic statistics (Section 3.1). Then, we compute drug embeddings and cluster them to investigate possible emerging patterns in the graph (Section 3.2).

### 3.1   Statistics

After we loaded the open drug knowledge graph into MySQL, we computed statistics of the coverage of each class across different sources. The results are shown in Table 2. Each source contributed a different profile of features, and some sources contributed unique classes. Specifically, Drugbank distinctly contributed the 'Manufacturer' and 'Interaction' classes while GoodRx contributed the 'Store' and 'Price' classes. Each source was linked back to Wikidata as a centralized source, based on the linkage methods described above. This shows the benefit of integrating sources with complementary foci in a single knowledge source, which is ultimately more than a sum of its parts.

### 3.2   Graph Embedding Analysis

We sought to further explore the higher-level structure of the extracted knowledge graph via graph embeddings. Our goal was to explore the embedding of the relation treatment (drug, condition) to confirm whether drugs that treat similar conditions are clustered together. If drugs are clustered in this fashion, the graph embeddings could enable drug recommendations, given a source drug, for providers in the future. Our embedding is built from all 3,654 instances in

| Class | Wikidata | WebMD | Drugbank | GoodRx |
|-------|----------|-------|----------|--------|
| Drug | 0 | 0 | 13573 | 0 |
| Product | 1560 | 1701 | 25637 | 0 |
| Condition | 900 | 568 | 0 | 0 |
| Treatment | 3654 | 977 | 0 | 0 |
| Interaction | 0 | 0 | 356254 | 0 |
| Manufacturer | 0 | 0 | 4092 | 0 |
| Store | 0 | 0 | 0 | 23 |
| Price | 0 | 0 | 0 | 20688 |

Table 2: Coverage of each class across different sources. Displays the total amount of instances of each entity within our data model, grouped by data source. Drugbank uniquely contributed to Interaction and Manufacturer features. GoodRx uniquely contributed to Store and Price features.

our treatment table, sourced from Wikidata. We then utilized the Ampligraph [6] and Tensorboard [1] libraries with TransE and [4] Complex [12] models to project our data into the 150-d embedding space. The training occurred for 200 epochs, with an Adam optimizer. A training set was generated from 90% of the data, with the remaining 10% set aside as testing data.

In Table 3, Our embedding models are evaluated using the following entity ranking tasks described by Wang et al [15]: 1) mean reciprocal rank (MRR), and 2) Hits@K. MRR asks the embedding model to rank unseen test triples. A model that produces higher ranks for known true triples (i.e. test triples) is considered superior at predicting missing links. The Hits@K metric computes how many elements of a vector of rankings make it to the top K positions. When visualizing the embedding vectors, we utilized embeddings from the Complex model as it performed best on our entity ranking tasks. To visualize the embedding, we reduced our embeddings into 3-d using T-SNE [7] as our dimensionality reduction method. We then inspected the result for nearest neighbors based on cosine similarity in the initial embedding space. In Figures 5 and 6, we selected results from our embedding visualization. The visualizations are from Tensorboard, and displayed using the aforementioned model parameters and visualization settings. In this embedding space, we found that drugs that treat similar conditions are somewhat clustered, while similar conditions are grouped together. For drugs, in Figure 5a, we find the 10 nearest cosine neighbors to source drug "insulin aspart". Two of the neighbors are also insulin variants, however, more domain expertise is required to deem whether this clustering is a meaningful representation of drugs that treat similar conditions. For conditions, in Figure 5b, the 10 nearest cosine neighbors are located for "bipoloar disorder". Many of the neighbors logically represent similar conditions such as "mood disorder", "schizophrenia", and "anxiety". Further experiments are required to confirm how meaningful these initial embeddings can be for recommending drug products. Other relations that may be helpful to include to achieve drug similarity embedding may be ICD-10 codes of the treatment's condition or the products of the treatment's drug. However,

these embeddings show early signs of progress for achieving goals around drug recommendation via nearest neighbor search within a graph embedding space.

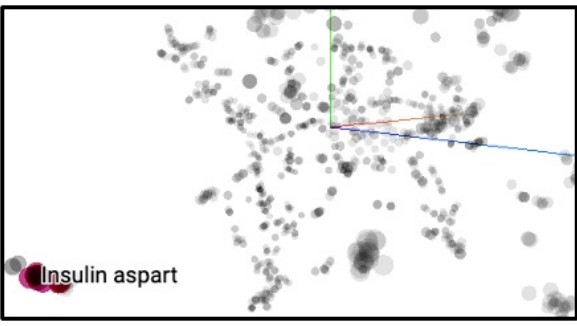

a. "Insulin aspart" neighbors

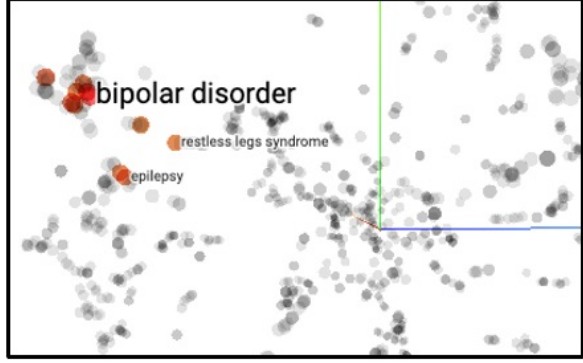

b. "Bipolar disorder" neighbors

Fig. 5: Graph Embedding Visualization. Visualization of all entities within the reduced embedding created by the Complex embedding model. In Figure 5a, the source entity 'insulin aspart' is selected. We observe clustering for this entity in the embedding space. In Figure 5b, 'bipolar disorder' is selected, which also exists within an observable cluster of similar entities.

## 4   Applications

In this Section, we present our web interface that allows user exploration of the relational data model. We also explore the associated property graph to gain motivation for future hypothesis and functionality Section 1.

| Drug: "Insulin aspart" | | Condition: "Bipolar disorder" | |
|---|---|---|---|
| Pramlintide | 0.059 | mood disorder | 0.309 |
| Insulin glargine | 0.098 | schizophrenia | 0.399 |
| Insulin human | 0.103 | insomnia | 0.401 |
| Insulin detemir | 0.109 | epilepsy | 0.428 |
| Metformin | 0.161 | psychosis | 0.433 |
| Insulin lispro | 0.175 | post-traumatic stress disorder | 0.446 |
| Exenatide | 0.231 | trigeminal neuralgia | 0.448 |
| Glyburide | 0.232 | anxiety | 0.460 |
| Glimepiride | 0.236 | restless legs syndrome | 0.468 |
| Acarbose | 0.242 | dysarthria | 0.481 |

Fig. 6: Embedding Nearest Neighbors by Cosine Distance. Nearest neighbors of the selected entities 'insulin asapart' and 'bipolar disorder'. Cosine similarity is the distance metric, based on entity vectors (d=150) in the original embedding space. We observe the embedding creates logically similar neighbors.

| Embedding | MRR | Hits@1 | Hits@3 | Hits@10 |
|---|---|---|---|---|
| TransE | 0.25 | 0.50 | 0.29 | 0.13 |
| Complex | 0.32 | 0.57 | 0.36 | 0.19 |

Table 3: Evaluation of Embedding Methods. Table 3 displays evaluation results of graph embedding models - TransE and Complex - using mean reciprocal rank (MRR) and Hits@k. Embedding models were created upon all 3,654 instances of the treatment(drug, condition) class.

## 4.1   Web Interface upon relational model

We prepare a web interface for our Intelligent Drug Shopper, shown in Figures 8 and 9. The web interface was developed using the Python Django framework. The user can input search parameters for a patient's condition, current medications, and price range. These parameters are inserted into a SQL query template that checks our data model for any matching results.

For example, in Figure 8 below, a patient is present with osteoarthritis and has a budget of 20 dollars to spend on medicine. These parameters are inputted and the query retrieves matching treatments, it's active ingredient, and average price. The user can then navigate to different views of the Active Ingredient or Product entity via hyperlinks.

To demonstrate further searching capabilities our data model provides, consider Figure 9. Extending the same search from Figure 8, a patient may also be taking some current medication like Zyvox, an antibiotic. This parameter is added to the search, and we find many of the previous recommended treatments

from Figure 8 are removed as they interact with this antibiotic. This feature will enable users to find treatments that avoid adverse drug interactions, while still treating a condition and adhering to the patient's budget.

Fig. 7: User Interface: Treatment Search by Price and Condition. In this example, a user is searching for product that treat the condition 'osteoarthritis' with a price between 0 and 20 dollars.

### 4.2   Visualizations upon graph model

In addition to exploration of the relational database via the web application, we also directed queries to the equivalent property graph stored in Neo4j. In Figure 7, we consider a user search for treatments of "medullary thyroid carcinoma" and all possible drug interactions with these treatments. The resulting visualization shows two possible treatments (cluster centers), with drug interactions branching outward. We can see there is an intersection of six drugs that interact with either treatment. Therefore, if a patient is currently prescribed a drug in this intersection, they cannot safely be prescribed either of the two treatments. Neo4j was utilized to perform this visualization. As the data model is loaded as a property graph in Neo4j, we can leverage Neo4j's wide set of graph analytics tools to compute such paths automatically. We also plan to use Neo4j to compute centrality metrics over our graph in the future.

## Intelligent Drug Assistant

### PRODUCT SEARCH

Condition: osteoarthritis

Price low: 0

Price high: 20

Current med: zyvox

Search

| Product | Active Ingredient | Average Price | Product Page |
|---------|-------------------|---------------|--------------|
| aspirin | Acetylsalicylic acid | 4.67 | aspirin |

Fig. 8: User Interface: Treatment Search by Current Medication. The data model enables the user to extend their search if they also input their current medication. For example, if the user is already taking medication 'Zyvox', many of the previous treatments can't be recommended as they interact with 'Zyvox'.

## 5    Discussion and future work

**Hub graph:** While we were able to link data sources with Wikidata, there are some benefits and drawbacks to the chosen design methodology. In our design, we link all sources back to Wikidata in a 'hub-and-spoke' fashion. No other sources are permitted to link to each other. This design functions to extend the Wikidata knowledge graph, enabling new drug features (e.g., drug price) to be analyzed with all other connected nodes to medication (Q12140). The drawback to this approach is that Wikidata does not contain nearly as many drugs or drug product entities as Drugbank, thus bottle-necking the amount of possible entity links made with other data sources. Depending on the application, the extension of Wikidata with drug-centric data may be less important. In this case, we would suggest using Drugbank as a centralized source for entity linkage to maximize the number of links on drug and drug product entities with other data sources.

   **Integration of more data sources**: To answer even more healthcare-centric questions, we propose to extend the knowledge graph with additional healthcare datasets. These datasets could relate to healthcare objects, such as prescriptions, procedures, diagnoses, claims, providers, payers, and healthcare facilities. Many of these datasets are made publicly available by government-run healthcare agencies, such as Food and Drug Administration (FDA[7]), National Institutes of Health (NIH[8]), and Center for Medicare Services (CMS[9]). Stan-

---

[7] https://www.fda.gov/home

[8] https://www.nih.gov/

[9] https://www.cms.gov/

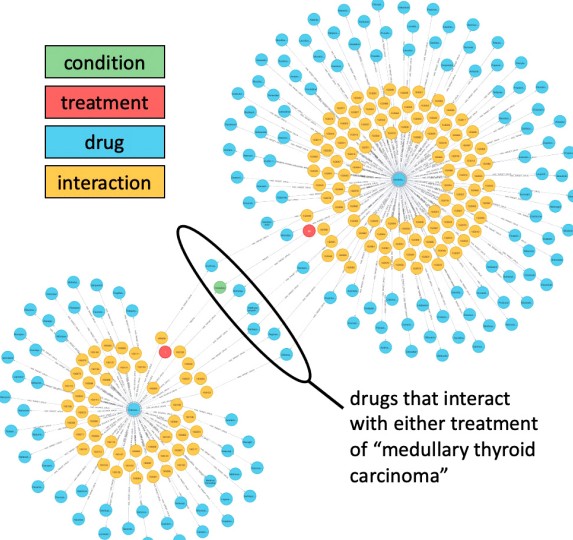

Fig. 9: Visualization: Treatment Interactions. Results of Neo4j query to find all treatments for 'medullary thyroid carcinoma' and all drug interactions for each treatment. For certain drugs, it may not be possible to avoid interactions with either treatment, evidenced by the intersection drug interactions for both possible treatments.

dard identifiers for each healthcare object is very common, and therefore can reduce the amount of fuzzy matching required to extend the knowledge graph. For example, the FDA gives drugs a National Drug Code (NDC), which is a representation of labeler, product, and package size. CMS gives each healthcare provider a National Provider Identifier (NPI). ICD-10 codes can be used to label medical conditions.

**Drug product similarity in embedding space**: A future hypothesis to check whether our knowledge graph can enable drug product similarity searching via kNN search within graph embeddings. This application would enable healthcare workers to find similar products for a source product. The embedding should be produced to cluster products together in an embedding space, if the products that treat similar conditions (ex: mental disorders, nervous, ocular). To enable this work, additional sources must be integrated - such as ICD(condition, ICD_code) - to enable ground truth checking of clusters.

**Application features**: Currently our application does not support searching based on multiple conditions or current medications. To support this, our query templates must be updated to allow for these additional search parameters. Another improvement would be to enable eager fuzzy n-gram searching, triggered by characters inputted in real-time, to find a matching indexed search term. This can be enabled via indexing of keywords and real-time searches upon the index.

This feature would enable higher success with user searches, compared to current functionality.

## 6   Conclusion

In this paper, we proposed the Open Drug Knowledge Graph: an integrated drug-centric data model used to enable customers to make well-informed purchasing decisions, by including prices, availability, and drug interactions in a single view without referencing fine print about drug-interactions. This data model leverages healthcare objects stored in pre-existing knowledge bases and integrates knowledge from previously disjoint systems. Our acquisition pipeline consists of three key steps: source data acquisition, entity resolution, and ontology mapping. When performing entity linkage, external drug identifiers such as Drugbank ID were heavily utilized, to reduce need for fuzzy matching. We created a web application to visualize the relational data model (MySQL), and showed its potential to be used by both healthcare workers and patients to inform treatment decisions. The model was also loaded into a property graph (Neo4j), which was anecdotally shown to enable visualization and network analytics upon the graph. We computed graph embeddings upon the treatment class using the TransE and Complex models. Nearest neighbor search based on cosine distance over these embeddings showed their potential to aid product and condition searches. We expect that such a single integrated source can help users make medically safe and financially smart decisions. Future work should investigate the usefulness of the graph for customers, but also integrate further sources and explore novel ways to leverage the data through graph centrality and path-finding methods. To facilitate further exploration and development of drug knowledge graphs, the data model[10] is made publicly available to the research community.

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
