# OpenReview forum: "Open Drug Knowledge Graph"
_eswc-conferences.org/ESWC/2021/Workshop/KGCW — KGCW 2021_

### Official Review · ~Giorgos_Flouris2 · 2021-04-09
**A decent contribution that deserves presentation in the workshop**

**Rating:** 7
**Confidence:** 4

**Review:**

This paper describes the merging (integration) of data from 4 different knowledge sources, pertaining to the domain of medical data (drugs, in particular). The work is very relevant to the workshop. Although not groundbreaking, I believe it provides a description of an interesting integration methodology that will be helpful for practitioners engaged in similar activities. The integrated data is represented in 3(!) different ways (relational, ontological, graph) and a visual interface that allows accessing the integrated data is described.

With regards to the paper's motivation, I have some concerns on the second bullet provided: in times of emergency, people will not have the time, or the composure, to look online for treatment. This is not meant to imply that the integration proposed by the authors is useless.

Section 2.1: it is not clear what the codes in parentheses are. After reading later sections of the paper, I would assume that they refer to IDs in the various knowledge sources; some clarification is necessary though.

I do not understand why the authors chose 3 equivalent, but different, representations of the integrated data (relational, ontological, graph). I would assume one is enough? If not, why?

Typos and other minor errors:
- "an total" (similar errors appear throughout the document, please check)
- Section 4, first paragraph: there is a strange reference to Section 1, probably a copy-paste remnant?

---

### Official Review · ~Samaneh_Jozashoori1 · 2021-04-12
**Open Drug Knowledge Base**

**Rating:** 5
**Confidence:** 4

**Review:**

This paper provides a drug knowledge base considering four heterogeneous sources including both scientific sources such as DrugBank and commercial ones e.g. GoodRx. This work tries to target **both** physicians and patients providing an integrated view of four drug-related databases for both medical e.g. drug side effects and non-medical e.g. price, information.

The motivation behind this work is clear and is in fact very important, impacting, and challenging topic.

The writing of the paper is fluent, however, there is a significant **lack of integrity** in the contents of the paper including the main **contributions**:
* In the **abstract** the main contribution of the paper is described as: a knowledge-base method to create a drug-centric knowledge **graph** which can have potentials (e.g. capturing drug similarity) based on their analysis. On the other hand, it is mentioned that the resulting knowledge **source** will be available in a relational database and a property graph.
* According to the **introduction** the main goal of the paper is to construct a knowledge **graph** for the user to provide an integrated view of the existing sources.
* In the implementation and application sessions, a **relational database** is created which is the base for the provided **web interface** for the users (not the knowledge graph). Also, a graph is created which is applied to perform analysis such as to observe if "the drugs that treat similar conditions" are clustered together.

Accordingly, the following questions and confusions will arise:
* Why knowledge **graph** while the user interface is provided on top of the relational database?
* If the answer to the above-mentioned question is the potentials that the graph will bring as mentioned in the embedding and visualization session, then why the **relational database**? Why the user interface is not on top of the knowledge graph?
* What are the specific differences between the schema of the provided relational database and the schema (ontology?) of the property graph from the semantic point of view?

I see the contributions of the paper to be clearly divided into two categories without a clear explanation of their correlations or complementarity in pursuing the main goal of the paper.

The paper claims to provide a knowledge base by **harmonizing** four sources, while except for entity resolution between Wikidata to WebMD, it applies DrugBank IDs in entity resolution and the entities will be discarded in case of not being matched by the IDs. **linking** may suit better in this context since **harmonization** requires further semantic level.

On page 7, it is mentioned that their ontology is described in figures 2 and 3, while figure2 is **Entity linkage metrics by data source** and figure3 is the **Relational schema**!?!

Minor issues and typos:
* Page 2, by "symptoms of drugs" do you mean "side effects of drugs"?
* Page4, last sentence of 2.1: "**A** total..."
* Page7, last sentence: "...corresponding **to** the..."
* Page8, Fig4 caption: "...format of **a** property..."

In summary, although the work is very interesting and important, I do not find the explained contributions integrated nor the paper ready to be published.

---

### Official Review · ~Pieter_Heyvaert1 · 2021-04-19
**Useful contribution but could be presented better**

**Rating:** 6
**Confidence:** 4

**Review:**

This paper introduces a knowledge graph (KG) about drugs, how they can treat symptoms, how they interact with each other and where they are available for purchase at what price.

I have a general remark regarding the use of the relation database and the KG. A lot of the text is talking solely about the database and how different methods are applied to its data. Is in these cases the KG not needed? Considering the paper's main contribution is the KG it is weird to read that most methods are only done on the database. When the KG is used, for example, to generate the graph in Figure 9, it feels like the KG is only there for visualization and not for the graph structure of the data itself.

It's not clear why Wikidata is chosen as the "hub", especially because the Drugbank is larger.

On page 7, it is not clear from Figure 3 how the projects goals (a, b, and c) are achieved. Also at the top of the page I think that the figures should be 3 and 4 instead of 2 and 3.

It is not clear how you go from Figure 3 to Figure 4 and how the latter still achieves the project's goals.

Is the code of the pipeline publicly available?

A lot of emphasis is put on the Web app, while it be more beneficial for the reader to have more details about the pipeline, considering that the knowledge graph, and how it is created, is the main topic/contribution of the paper.

Grammar/typos
- The opening quotes seem to be facing the wrong way.
- Page 9: "In Table 3, Our" --> "In Table 3, our"
- Page 11: section 4.1 "it's active ingredient" --> "its active ingredient"
- Page 12 section 4.2: "Figure 7" --> "Figure 9"

---

### Meta-Review · Program_Chairs · 2021-04-21

**Recommendation:** Accept
**Confidence:** 5

**Metareview:**

The three reviewers agreed on the relevance of the paper for its presentation in the workshop. But please, take into consideration the comments they've provided, mostly focused on the improvements about the presentation of the ideas and integration of the paper in a unique story, for the camera-ready version. It would be also good to emphasize how the KG has been created (as it is the main contribution of the work and also the principal focus of the workshop, which would create some discussion during the paper presentation) instead of the ideas on the development of a webapp, which is not very relevant from a research perspective.

David

---

### Decision · Program_Chairs · 2021-04-23

Accept